# Careful design of Large Language Model pipelines enables expert-level retrieval of evidence-based information from syntheses and databases

Radhika Iyer[1], Alec Philip Christie [1,2]*, Anil Madhavapeddy[3], Sam Reynolds[1], William Sutherland[1], Sadiq Jaffer[3]

1 Department of Zoology, University of Cambridge, Cambridge United Kingdom, 2 Centre for Environmental Policy, Imperial College London, United Kingdom. 3 Department of Computer Science & Technology, University of Cambridge, Cambridge United Kingdom,

* a.christie@imperial.ac.uk

## Abstract

Wise use of evidence to support efficient conservation action is key to tackling biodiversity loss with limited time and resources. Evidence syntheses provide key recommendations for conservation decision-makers by assessing and summarising evidence, but are not always easy to access, digest, and use. Recent advances in Large Language Models (LLMs) present both opportunities and risks in enabling faster and more intuitive systems to access evidence syntheses and databases. Such systems for natural language search and open-ended evidence-based responses are pipelines comprising many components. Most critical of these components are the LLM used and how evidence is retrieved from the database. We evaluate the performance of ten LLMs across six different database retrieval strategies against human experts in answering synthetic multiple-choice question exams on the effects of conservation interventions using the Conservation Evidence database. We found that LLM performance was comparable with human experts over 45 filtered questions, both in correctly answering them and retrieving the document used to generate them. Across 1867 unfiltered questions, LLM performance demonstrated a level of conservation-specific knowledge, but this varied across topic areas. A hybrid retrieval strategy that combines keywords and vector embeddings performed best by a substantial margin. We also tested against a state-of-the-art previous generation LLM which was outperformed by all ten current models – including smaller, cheaper models. Our findings suggest that, with careful domain-specific design, LLMs could potentially be powerful tools for enabling expert-level use of evidence syntheses and databases in different disciplines. However, general LLMs used 'out-of-the-box' are likely to perform poorly and misinform decision-makers. By establishing that LLMs exhibit comparable performance with human synthesis experts on providing restricted

**Data availability statement:** Data and code associated with this paper can be found at: https://doi.org/10.5281/zenodo.14049626.

**Funding:** RI was supported by a UROP internship at the University of Cambridge and an unrestricted donation from Tarides. SJ was supported by an unrestricted donation from John Bernstein. APC received financial support from Imperial College London through an Imperial College Research Fellowship grant, as well as a Henslow Fellowship funded by the Cambridge Philosophical Society. The funders had no role in study design, data collection and analysis, decision to publish, or preparation of the manuscript.

**Competing interests:** The authors have declared that no competing interests exist.

responses to queries of evidence syntheses and databases, future work can build on our approach to quantify LLM performance in providing open-ended responses.

## Introduction

To maximise our chances of bending the curve of biodiversity loss [1] with limited time and resources, we need to use the best available evidence to increase the effectiveness of conservation efforts whilst also reducing harmful impacts to both nature and local communities [2–5]. Whilst evidence syntheses are key to informing more effective practice and policy by distilling the scientific literature [6–8], they are not always easy for decision-makers to access, digest and use [9–11]. However, recent advances in the capabilities of Large Language Models (LLMs), statistical models typically optimised to predict and generate text that should appear next in a sequence, have made it easier than ever to gain access to summaries of information and on-demand answers to specific questions across a variety of domains [12–15]. LLMs clearly have considerable potential to summarise information efficiently from vast corpuses of literature, but their use in rigorously supporting effective decision-making remains controversial and fraught with potential pitfalls and dangers, including the risk of misinformation from biases and errors [16,17].

There is particular concern that the proliferation of natural language interfaces or 'chatbots' (e.g., OpenAI's ChatGPT, Microsoft Co-Pilot, and Google's Gemini) could misinform decision-makers who may be attracted to LLMs for their ability to rapidly and flexibly answer their questions [20–25], particularly in crisis disciplines such as biodiversity conservation [26]. LLM usage by those providing advice to decision-makers (e.g., consultants and scientists) could also further compound poor quality decision-making without properly considering the quality and relevance of evidence [16,23,27,28]. For example, LLMs used 'out-of-the-box' have the potential for hallucinations, false references and citations, out-of-date and biased information based on inherent biases in the training data used to build these models [29]. It is therefore vital to determine whether reliable implementations of LLMs that derive their information from robust evidence syntheses and databases can be developed that enable rapid access to evidence bases without misinforming decision-makers.

Few studies have quantitatively evaluated the reliability of using LLMs for decision support [12,20] and none, to our knowledge, in conservation have made comparisons with human experts. It is also unclear how recent advances in fine-tuning and text retrieval techniques (e.g., Retrieval Augmented Generation: RAG; Box 1), designed to improve the performance of LLMs in specific contexts and reduce errors, may help to minimise the risks of misinforming decision-makers [30,31]. We also lack an understanding of the subject-specific information that different LLMs possess across different topic areas and therefore whether certain LLMs might provide more reliable decision support for certain fields.

Box 1. Glossary of terms used in this study

- **Large Language Models** (LLMs) are neural networks trained on large quantities of text. Given a block of text, called the *context*, they are optimised to predict the small piece of text that follows. Adding this small piece of text to the context and repeating the process enables text generation. With tuning, these models can follow instructions and carry out multi-turn conversations.

- **Context** is the initial set of text given to a LLM that contains instructions and potentially data for it to use. LLM responses to long contexts are expensive (resources scale quadratically with size of the context or 'context window' for most state-of-the-art models) and so techniques have been developed to focus on only including relevant information in the context.

- **Retrieval-Augmented Generation** (RAG) is one such technique for identifying relevant information to populate the context. In this approach, a user's query is used to *retrieve* potentially relevant documents from a corpus, and these are then added to the context along with the query and sent to the LLM.

- **Retrieval strategies** are methods for finding relevant information from a corpus:- **Sparse retrieval** uses keyword-based metrics, which can work well when there is significant keyword overlap between queries and documents, but can fail when semantically related documents do not share keywords.

- **Dense retrieval** uses a small language model to *embed* documents (or document chunks) as datapoints in a high dimensional space and is optimised to minimise the distance of semantically related text in the embedding space. The user query is embedded and the closest documents in the corpus are returned as relevant.

- **Hybrid retrieval** using a small language model called a *reranker* to compare documents retrieved from **dense** and **sparse retrieval** methods with the query. This can increase performance as the reranker model has access to the full text of potentially relevant documents.

- The **Conservation Evidence database** [18,19] contains over 8858 studies, at time of writing, that test the effectiveness of conservation actions on biodiversity outcomes – currently 3890 conservation actions have been identified, of which 2,250 are tested by a study. Summaries of the evidence for each conservation action are organised into topic areas called synopses (e.g., Amphibian, Peatland, and Bird Conservation). For each action, key message paragraphs summarise study findings and background paragraphs provide contextual information on the action, whilst an effectiveness category (derived from structured expert elicitation) gives a likely indication of the benefits and/or harms of the action whilst accounting for the certainty of the evidence.

Before evaluating how LLMs perform at providing open-ended, free-text responses to questions, we first need to understand their performance at providing restricted responses to questions and retrieving the correct information. We expect LLM performance to decline as tasks become more complex and so we first need to establish where performance issues occur. Once a performance threshold is determined, we can design systems accordingly to ensure the risk of misinforming decision-makers is minimised, with comparable or lower levels of error than humans. Here we begin this process by first testing LLM performance on relatively simple tasks with restricted responses.

In this study, we ask ten different state-of-the-art LLMs to provide answers to conservation-related multiple-choice questions, including identifying the relevant source document used to answer each question (Fig 1). We use the Conservation Evidence database as a case study (Box 1), asking LLMs and human experts (who compiled the database) questions on the effects of conservation interventions for which evidence exists in this database (Fig 1). We adapt and apply an automated method [32] to measure the task-specific accuracy of Retrieval-Augmented LLMs using automatically

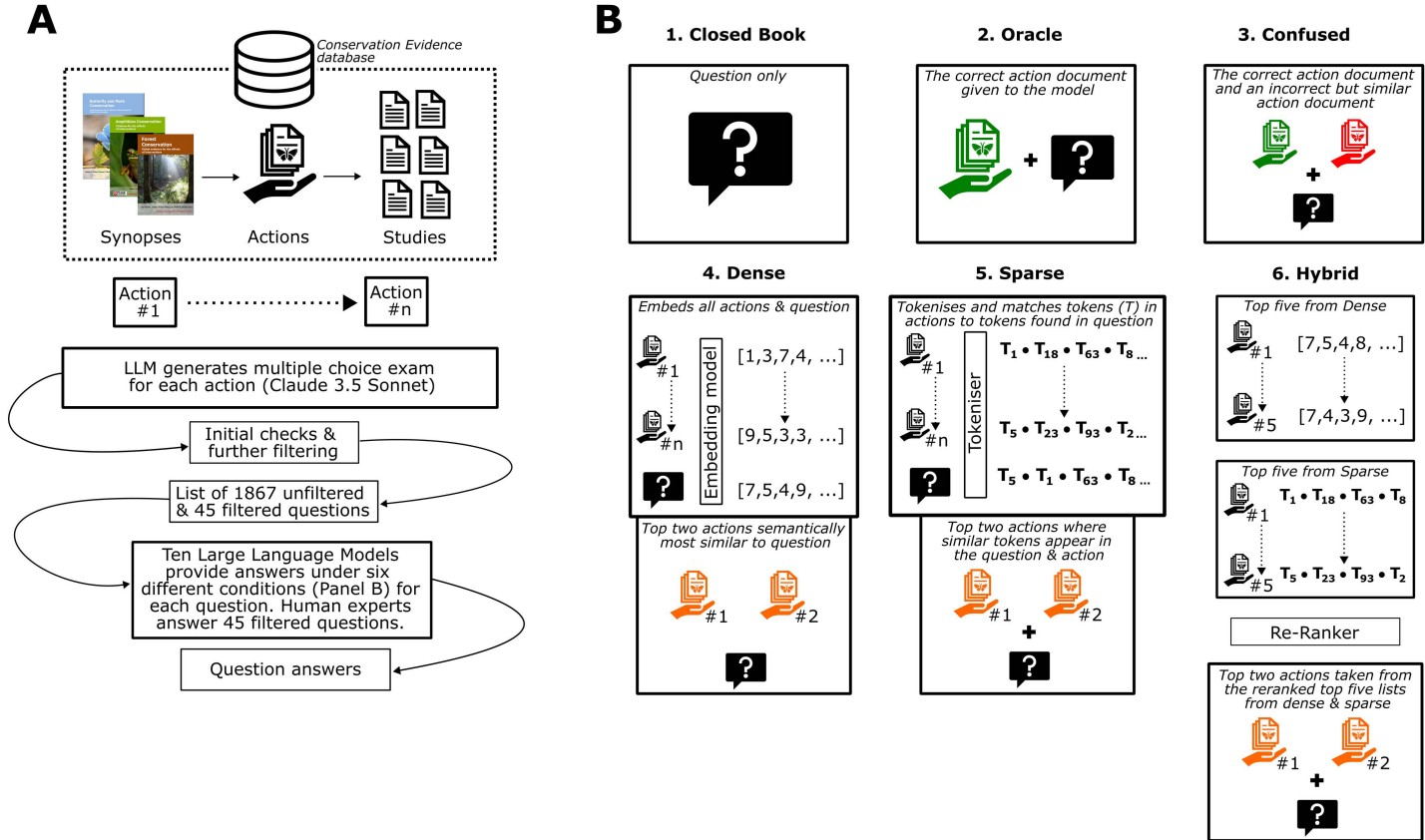

**Fig 1. A Large Language Model (LLM), Claude 3.5 Sonnet, was used to generate a multiple-choice exam for each of the 2,250 actions using an automated method[30], (Panel A and B) excluding questions that solely asked questions based on the effectiveness categories in the Conservation Evidence database.** This formed a larger set of 1867 unfiltered questions. We also refined these down to a filtered set of 45 questions to enable a comparison with human experts, ensuring that filtered questions were clear and could be answered with a single, accurate answer. Ten LLMs were then asked to provide answers under six different exam conditions for each question within the unfiltered and filtered sets (Panel **B)**. These exam conditions included three different types of retrieval strategies: sparse, dense, and hybrid retrieval (Box 1).

generated synthetic exams composed of multiple-choice questions based on a given corpus of documents (Fig 1; Methods). We generate a large unfiltered dataset of questions to ask LLMs to gain a broad overview of their performance using different document retrieval strategies and on different conservation topics, as well as a smaller, filtered set of questions to conduct a comparative evaluation between LLMs and human experts (Fig 1). These evaluations represent a cost-efficient, interpretable, and robust strategy to select optimal components for an initial RAG system that can provide an intelligent search function for users of the Conservation Evidence database.

To determine the robustness of LLMs to errors, we ask whether each LLM performs better at answering questions and retrieving the correct source document (containing the answer) compared to: 1. random guessing; 2. human experts that curated the Conservation Evidence database; 3. other LLMs; and 4. a predecessor of one of the LLMs - i.e., to see whether there has been improvement over time. Our approach could be applied across fields and disciplines to evaluate the suitability of different LLMs and retrieval strategies for providing decision support on specific subjects.

In the next section, we document our study's methodological approach, including information about the Conservation Evidence database that we used as the basis of the study, how we generated and filtered synthetic exam questions for LLMs and human experts, and how we conducted our statistical analyses. We then describe our results, first in terms of

comparing the performance of LLMs and human experts on a smaller, filtered dataset of questions. This is followed by comparing how different LLMs and retrieval strategies performed against each other and across different fields of conservation on a larger dataset of questions. Finally, we discuss our key findings, the limitations of our current study and future research directions, as well as the potential risks and opportunities posed by using LLMs for decision support using evidence databases.

## Materials and methods

We evaluated the performance of different Large Language Models (LLMs) in answering questions based on the evidence contained within the Conservation Evidence database as a case study. Our methodology was adapted from a previously published automated approach [32], which is designed to measure the task-specific accuracy of Retrieval-Augmented Generation (RAG) LLMs using automatically generated synthetic exams composed of multiple-choice questions based on a given corpus of documents (Fig 1).

### Conservation Evidence corpus

The Conservation Evidence database comprises a collation of studies testing the effects of conservation interventions on biodiversity outcomes for 26 topic areas (called synopses) structured around various species and habitat groups (e.g., Birds, Bees, Terrestrial Mammals). Each synopsis contains a range of actions (22–400 per synopsis), with each action including an action title and number, background information, key messages, and evidence summaries – this text constituted a 'document' for the purposes of our study (S1 Fig in S2 File), if any evidence has been found to test that action. For this study, we focused on the 2,250 actions that had at least one associated study. See S1 Fig and S2 Fig in S2 File for examples of summaries of the evidence for different actions.

### Exam generation

We used Claude-3.5 Sonnet for exam question generation, as it was the strongest publicly available LLM at the time of the study. The exam generation process involved several steps:

1. **Corpus Preparation**: We reduced the action documents to key messages and removed actions without supporting evidence.

2. **Question Generation**: We generated 3–4 questions per action using Claude-3.5 Sonnet (see S1 Table in S2 File for prompt).

3. **Filtering Process**: We applied several filters to ensure question quality:

   - Shuffled answer options to prevent bias.

   - Removed questions referring directly to actions (for closed book testing).

   - Applied Jaccard similarity thresholds to ensure we generated high quality incorrect answers (or discriminators [32]) – i.e., we removed questions that contained multiple rephrased correct answers (intra-candidate similarity), or where the phrasing of the correct answer reflected that of the question, giving away the answer (extra-candidate similarity).

   - Removed questions solely based on effectiveness ratings from the CE database (e.g., "How effective is..").

We deviated from an automated evaluation method[30] by not applying Item Response Theory for iterative question filtering due to using all post-filtering questions across evaluated LLMs. This gave a large dataset of 1867 unfiltered questions, which whilst potentially containing inaccuracies, provided a basis for comparing relative performance across different LLMs, retrieval strategies, and conservation topic areas (Fig 1).

## Human evaluation

We also curated a filtered dataset of 45 questions (checked for clarity and accuracy – see below) to measure the absolute performance of different LLMs and retrieval strategies and provide a human evaluation benchmark. To produce this filtered dataset, we:

1. Randomly sampled three actions from each synopsis.

2. Applied automated filtering as described above.

3. Iterated the process to ensure each synopsis had at least one question.

4. Manually filtered the questions down to a final subset of 45 questions by two reviewers, who were researchers from the Conservation Evidence project team and did not take part in answering the survey questions. The filtering was based on three main criteria: 1. the clarity of the question, 2. the accuracy of the answers, and 3. that there was one clear correct answer. Both reviewers independently selected questions to reject and then met to discuss any disagreements. The final subset of 45 questions were then reviewed again by both researchers to check they met all three criteria.

To establish a human evaluation benchmark, we formed a human expert comparison group that represented the Conservation Evidence project team (six experts, after excluding two experts who filtered the questions - see above) based at the University of Cambridge. These experts were uniquely qualified to answer questions and retrieve information on the Conservation Evidence database. Their experience in curating and using the online database [18,19] to retrieve information to answer practitioner questions on conservation interventions provided the ideal human comparison for this evaluation. This human expert group set a benchmark representing highly expert use and querying of the database with which to compare LLMs and retrieval strategies against, which could not be achieved by sourcing experts from a wider group of conservation experts.

The recruitment period for participants ran from 26th August 2024–25th October 2024. We obtained written informed consent via series of questions in a Qualtrics survey form that all participants had to complete before answering the multiple choice questions for this study. Please see Participant Information Sheet and Consent Forms in the Supporting Information for more details. All participants' data were also anonymised. We obtained ethical approval from the University of Cambridge Computing Science Ethics Committee (review no. #2324).

Each human expert answered all 45 multiple choice questions using an online anonymous survey implemented via Qualtrics survey software. The questions recorded the responses of these participants to these questions and asked them to provide a link to the action webpage (source document) that they used to answer the question (i.e., where they retrieved the evidence from). We also timed how long it took each participant to answer each question using a built-in timing function in Qualtrics survey software. We used the median time per question and its interquartile range as conservative measures as a small number of extreme values were generated, probably caused by participants taking breaks during questions.

## Retrieval strategies

We evaluated three retrieval strategies (Box 1) that are commonly used in RAG systems to retrieve relevant information:

1. **Dense Retrieval**: We used Nomic Embed Text v1.5 through the SentenceTransformers (SBERT) module for embedding. This method represents documents and queries as dense vectors in a high-dimensional space, allowing for semantic similarity comparisons.

2. **Sparse Retrieval**: We employed BM25, a probabilistic retrieval function that ranks documents based on the appearance of query terms, considering term frequency and document length.

3. **Hybrid Retrieval**: We used a cross-encoder (ms-marco-MiniLM-L-6-v2) for re-ranking. This approach combines the strengths of both dense and sparse retrieval, using an initial retrieval step followed by a more computationally intensive re-ranking step.

All retrieval strategies selected two action source documents that were added to the prompt sent to the LLM (S1 Table in S2 File).

**LLM evaluation**

We evaluated the following state-of-the-art LLMs at the time of conducting the study:

- Llama 3.1 8B Instruct-Turbo (FP8)

- Llama 3.1 70B Instruct-Turbo (FP8)

- Gemma2 Instruct - 9B (BF16)

- Gemma2 Instruct - 27B (BF16)

- Mixtral 8x22B Instruct (BF16)

- Gemini 1.5 Flash (gemini-1.5-flash-001)

- Gemini 1.5 Pro (gemini-1.5-pro-001)

- Claude 3.5 Sonnet (claude-3-5-sonnet-20240620)

- GPT-4o (gpt-4o-2024-08-06)

- GPT-4o mini (gpt-4o-mini-2024-07-18).

We used Google Cloud's Vertex AI for Gemini 1.5 Pro and Gemini 1.5 Flash, OpenAI for GPT-4o and GPT4o mini, Claude for Claude 3.5 Sonnet and Deepinfra for all other models. To maximise the repeatability of LLM responses, we set the temperature hyperparameter to the minimum value of 0 – this effectively minimises the randomness of LLM responses and maximises predictability as much as possible (i.e., almost deterministic). See S1 Table in S2 File for the prompt structure used and examples of prompts. We tested these LLMs under six different exam conditions (Fig 1) reflecting different retrieval strategies (or lack of them):

1. Closed book (no source documents provided). This scenario represents the inherent knowledge encoded in the weights of the LLM (also called its parametric or 'base' knowledge; [32]).

2. Oracle (using the singular source document used to generate the question). This scenario provides the LLM with access to the ground truth information.

3. Confused (oracle document plus one random document from an action with evidence in the same synopsis). This scenario provides the LLM with access to the ground truth information, like the oracle, but adds a potentially confusing document as extra information.

4. Dense retrieval (two documents retrieved via a dense model – see Retrieval Strategies section or Box 1).

5. Sparse retrieval (two documents retrieved via a sparse model – see Retrieval Strategies section or Box 1).

6. Hybrid retrieval (top 5 documents via dense retrieval added to top 5 documents via sparse retrieval, which are reranked to two top documents; Box 1).

Closed book and oracle scenarios act as lower and upper bounds, respectively, on the quality of the information that an LLM can be served with from the corpus [32]. The confused scenario acts as an upper bound on the realistic performance

of an LLM because a retrieval strategy must be used to work out how to serve the LLM with the correct source document (rather than just being given it in the oracle scenario). The 'confusion' document is so-called as it is unlikely to provide useful information to help the LLM answer the question. This is because the questions being asked related to a specific conservation action, given the structure of the Conservation Evidence database, and an action document is a written summary of the evidence for a discrete conservation action. Therefore, the confused scenario helped us to assess whether an additional unrelated document decreases performance or not (i.e., is performance for confused worse than oracle?). We also investigated the difference between the three retrieval strategy scenarios (dense, sparse, and hybrid) and the confused scenario, since this will reflect how well they can serve the LLM with the correct source document (unlike confused and oracle that guarantee this).

## Statistical testing

First, we tested the null hypotheses that there was no difference in the accuracy of each LLM (using a hybrid retrieval strategy) and a random guesser (with a 25% chance) at correctly answering the 45 multiple-choice questions. We used a modified permutation test, with a similar approach to a sign test, whereby we directly compared the correct and incorrect answers given by each LLM versus a random guesser. For questions, where the random guesser gave a correct answer and the LLM gave an incorrect answer, a value of -1 was assigned. For the converse, a value of $+1$ was assigned (i.e., positive values = LLM wins, negative values = random guesser wins). For cases where both the LLM and random guesser got the questions right or wrong (i.e., a draw), a value of 0 was assigned. As there was a 25% chance of there being a draw for each question, we subtracted the number of draws that might have occurred by chance across the 45 questions – this was done by randomly sampling from the numbers 0 and 1 (0 = no draw, 1 = draw, with a probability of 0.75 of no draw and 0.25 of a draw) with replacement 45 times and taking the sum. We calculated a test statistic by taking the sum of the -1s, $+1$s, and remaining 0s across the 45 questions as in a conventional sign test. This process was repeated 10,000 times to produce a raw p-value by calculating the proportion of test statistics equal to zero. Since we tested ten null hypotheses (one for each LLM) separately, we used the Holm adjustment to correct p-values for multiplicity. We also calculated the mean, standard deviation, and 95% Confidence Intervals of the permutation test statistic.

Second, we repeated the permutation test described previously to test the null hypotheses that there was no difference in the accuracy of each LLM and a randomly selected human expert at answering multiple-choice questions. This approach enabled us to account for: 1. the probability that a human expert and LLM may give the same answer by chance based on the four possible answers to the multiple choice question; and 2. the fact we were interested in draws, cases where LLMs outperform a human expert and vice versa. We used paired data for both six human experts and each LLM (using a hybrid retrieval strategy) for all the 45 filtered questions. For each question, we randomly selected one of the six human expert's answers (allowing us to effectively bootstrap our sample and test our null hypothesis directly), comparing its correctness to the correctness of the given LLM's answer. This process was repeated 1,000,000 times given the large number ($6^{45}$) of possible combinations to ensure we obtained robust test statistics, associated error, and p-values. Our bootstrapping approach helped to improve the robustness of our human-LLM comparison, given that the expert group (with intimate knowledge of the Conservation Evidence database) that we required for this evaluation was relatively small and unsuitable for more conventional tests such as McNemar's test.

Third, we repeated the permutation test for the null hypotheses that there was no difference in the retrieval accuracy of each of the three retrieval strategies (sparse, dense, and hybrid) and a randomly selected human expert (for the same reasons as before). However, as the number of possible retrieval answers was large (2250 potential action pages to retrieve), we did not need to adjust for the probability of draws by chance as we did previously for the multiple-choice answers. For each question, we randomly selected one of the six human expert's retrieval responses, comparing its correctness to the correctness of the given retrieval strategy's response. This process was repeated 1,000,000 times given the large number ($6^{45}$) of possible combinations.

Finally, we used two separate logistic regression Generalised Linear Models (GLMs) to test the null hypotheses that there were no differences in 1.) the accuracy of different LLMs at answering questions; and 2.) the retrieval accuracy of sparse, dense, and hybrid retrieval strategies. GLM selection was carried out via likelihood ratio tests of nested GLMs using the 'anova' function and Aikake Information Criterion corrected for small sample sizes (threshold of > 2 ΔAICc). Our initial GLM consisted of two interaction terms: LLM accuracy ~ Synopsis*LLM + LLM*Exam Type to test whether certain LLMs performed better for certain synopses, and whether certain LLMs performed better for certain types of exams (e.g., closed book, oracle, hybrid retrieval, etc.). The best GLM selected contained all the explanatory variables but without any interactions: LLM accuracy ~ Synopsis + LLM + Exam Type. GLMs for each possible combination of variables were also tested, including a null intercept-only GLM. For the second GLM, we initially started with an interaction term: Retrieval accuracy ~ Synopsis*Retrieval Strategy, but the best GLM selected contained both variables but without an interaction (Retrieval accuracy ~ Synopsis + Retrieval Strategy). GLMs for each possible combination of variables were also tested, including a null intercept-only GLM.

Statistical significance of covariates in the best GLMs were assessed with an analysis of deviance test (Type II) using the 'Anova' function from the R package car (Fox and Weisberg 2019). We conducted pairwise tests between the levels of the categorical explanatory variables using the emmeans package with a Tukey adjustment for multiplicity (Lenth 2021).

## Results

### Comparison with human experts for filtered dataset

**Open book performance of LLMs demonstrates competitiveness with human expert performance.** Across all 45 selected questions, three LLMs, GPT-4o (97.8%), Llama 3.1 70B (97.8%), and Gemma 2 27B (95.6%) had a higher raw average performance than human experts (mean = 94.8%; median = 95.6%; IQR = 93.8–95.6%; Range = 91.1%-97.8%) when provided access to the Conservation Evidence corpus and using a hybrid retrieval strategy, but these differences were not statistically significant (see below and Table 1). All LLMs performed significantly better than a random guesser (means of 27.8–32.8 more correct answers versus a random guesser; p = 0 for all LLMs; S2 Table in S2 File), reflecting the fact that the LLM percentage accuracy was substantially higher than the 25% expected by chance.

Most LLMs had a comparable level of performance at correctly answering questions to the average human expert (Table 1). Mean test statistic values were typically small with narrow 95% Confidence Intervals (CIs) that overlapped with zero, indicative of the large number of draws observed in our datasets (beyond those expected by chance) and thus the highly similar performance of each LLM and a randomly selected human across the 45 questions (Table 1). Therefore, for most LLMs, we could not reject the null hypothesis (p > 0.05; Table 1) that there was no difference in the head-to-head performance of the LLM and a randomly selected human expert, except for Llama 3.1 8B Instruct Turbo which performed significantly worse than the human experts (mean = 3.7 fewer correct answers; 95% CIs = 2–5 fewer correct answers; p = 0.008; Table 1). Gemma2 - 9B Instruct also appeared to perform worse than a random human expert (mean = 2.7 fewer correct answers; 95% CIs = 1–4 fewer correct answers; Table 1), but we could not reject the null hypothesis at the 0.05 significance level after adjusting p-values for multiple comparisons. Conversely, GPT-4o and Llama 3.1 70B Instruct Turbo almost always matched or exceeded human experts (mean = 1.3 more correct answers; 95% CIs = 0–3 more correct answers; Table 1).

**Comparison of retrieval strategy performance with expert retrieval.** The hybrid strategy's overall retrieval accuracy (i.e., identifying the correct source document to answer the question) was 88.9%, narrowly outperforming the raw average of human experts (mean = 87.8%; median = 88.9%; IQR = 85.6–88.9%; Range = 82.2–93.3%).

Using equivalent permutation tests as before, we found that the per-question retrieval accuracy of dense and sparse retrieval strategies was significantly worse than a randomly selected human expert (Table 1 – Dense: mean = 3.5 fewer correct retrievals, 95% CIs = 1–6 fewer correct retrievals, p = 0.029; Sparse: mean = 7.5 fewer correct retrievals, 95%

**Table 1. Results of a paired comparison of question accuracy of Large Language Models using a hybrid retrieval strategy versus human experts, as well as comparing retrieval accuracy of different retrieval strategies versus human experts for the filtered 45-question set.** Test statistics are from a permutation test used to test two null hypotheses: 1. LLM: no difference in multiple-choice question accuracy of the given LLM and a randomly selected human expert; 2. Retrieval strategies: no difference in retrieval accuracy of a retrieval strategy and a randomly selected human expert. The test statistic can be interpreted as follows: negative values = random guesser or randomly selected human expert answered more questions correctly; positive values = LLM/retrieval strategy answered more questions correctly, and zero = all draws or equal numbers of wins and losses. The test statistic accounts for the number of draws expected by chance for the multiple-choice questions. *denotes statistically significant Holm-adjusted p-value at 0.05 significance level.

| Large Language Model | Overall percentage question accuracy | Holm-adjusted p-value | Mean permutation test statistic | SD permutation test statistic | Lower 95% CI | Upper 95% CI |
|---|---|---|---|---|---|---|
| GPT-4o | 97.80% | 1.000 | 1.331 | 1.000 | 0.000 | 3.000 |
| Llama 3.1 70B Instruct Turbo | 97.80% | 1.000 | 1.335 | 1.000 | 0.000 | 3.000 |
| Gemma2 - 27B Instruct | 95.60% | 1.000 | 0.333 | 0.999 | -1.000 | 2.000 |
| GPT-4o Mini | 93.30% | 1.000 | -0.667 | 1.000 | -2.000 | 1.000 |
| Mixtral 8x22B | 93.30% | 1.000 | -0.667 | 1.000 | -2.000 | 1.000 |
| Claude 3.5 Sonnet | 93.30% | 1.000 | -0.665 | 1.001 | -2.000 | 1.000 |
| Gemini 1.5 Flash | 91.10% | 0.801 | -1.667 | 1.000 | -3.000 | 0.000 |
| Gemini 1.5 Pro | 91.10% | 0.801 | -1.668 | 1.000 | -3.000 | 0.000 |
| Gemma2 - 9B Instruct | 88.90% | 0.135 | -2.666 | 1.000 | -4.000 | -1.000 |
| Llama 3.1 8B Instruct Turbo | 86.70% | 0.008* | -3.668 | 1.000 | -5.000 | -2.000 |
| **Retrieval strategy** | **Overall percentage retrieval accuracy** | **Holm-adjusted p-value** | **Mean permutation test statistic** | **SD permutation test statistic** | **Lower 95% CI** | **Upper 95% CI** |
| Dense | 80.0% | 0.029* | -3.500 | 1.404 | -6.000 | -1.000 |
| Sparse | 71.1% | 0.000* | -7.501 | 1.407 | -10.000 | -5.000 |
| Hybrid | 88.9% | 0.270 | 0.500 | 1.404 | -2.000 | 3.000 |

CIs = 5–10 fewer correct retrievals, p < 0.001). However, we could not reject the null hypothesis (of no difference in retrieval accuracy versus a random expert) for the hybrid retrieval strategy and found comparable levels of accuracy with human experts (mean = 0.5 more correct retrievals, 95% CIs = 2 fewer to 3 more correct retrievals; p = 0.271; Table 1). This was again associated with large numbers of draws between human experts and the hybrid retrieval strategy.

The median time taken to answer each question by a human expert was 139.5 seconds (IQR 90.9–269.1 seconds) versus practically instantaneous responses from any LLM, given retrieval had already been completed and documents effectively cached.

## Model comparisons with unfiltered dataset

**Question accuracy across synopses, exam conditions, and retrieval strategies.** We found that for the larger dataset of questions, LLM accuracy differed significantly between synopses, LLMs, and exam conditions (S3 Table in S2 File, S4 Table in S2 File). The accuracy of LLMs was significantly higher for the Biodiversity of Marine Artificial Structures and Natural Pest Control synopses than almost all other synopses – differences were non-significant with each other and the Bat and Bee synopses (S5 Table in S2 File, Fig 2). Synopses that LLMs performed worst for included Reptile Conservation, Control of Freshwater Invasives, and Butterfly and Moth Conservation (Fig 2; S4 Table in S2 File, S5 Table in S2 File).

Generally, LLMs had significantly higher accuracy under oracle conditions than all other exam types, followed by confused, hybrid retrieval, dense retrieval, sparse retrieval, and closed book in descending order (Fig. 2; S3 Fig in S2 File) – all differences were statistically significant (S6 Table in S2 File).

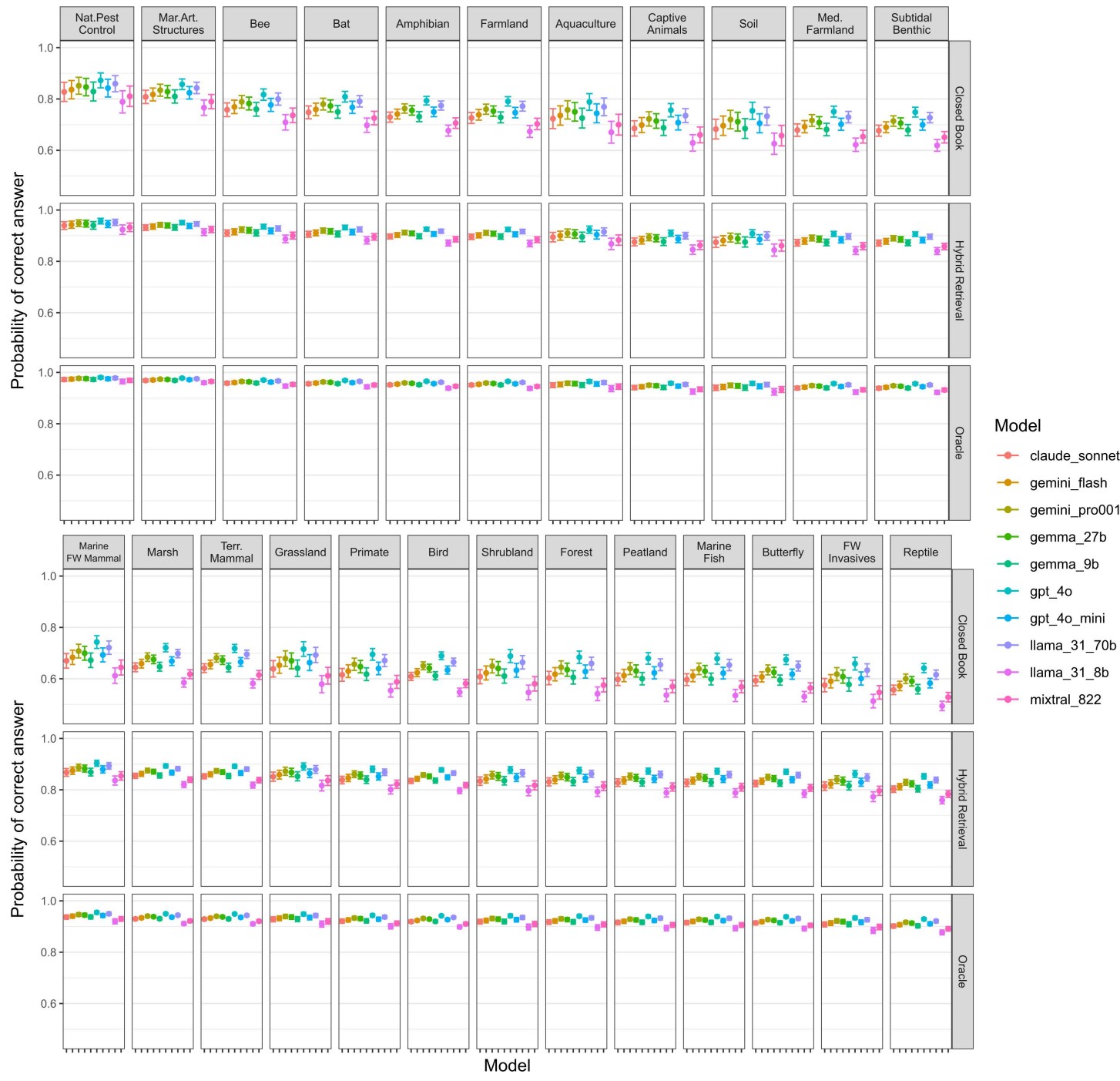

**Fig 2. Logistic regression Generalised Linear Model (GLM) predictions of the accuracy of LLMs across different synopses, under different exam types (mean and 95% Confidence Intervals).** The results for confused, sparse, and dense retrieval are found in S3 Fig in S2 File.

**Table 2. Overall Large Language Model (LLM) accuracy across different exam conditions on the unfiltered dataset.** The table is sorted by LLM performance under the hybrid retrieval strategy. Results for the filtered 45-question dataset are presented in S8 Table in S2 File. We also specify whether LLMs are open or closed source.

| LLM | Closed Book | Oracle | Confused | Dense | Sparse | Hybrid | Open/closed source |
|---|---|---|---|---|---|---|---|
| GPT-4o | 69.8% | 95.3% | 94.4% | 86.1% | 83.2% | 90.2% | Closed |
| Llama 3.1 70B Instruct Turbo | 68.7% | 94.3% | 92.7% | 85.3% | 82.5% | 87.9% | Open |
| Gemini 1.5 Pro | 65.5% | 94.9% | 92.8% | 84.4% | 81.3% | 87.9% | Closed |
| Gemma2 - 27B Instruct | 66.1% | 93.7% | 92.3% | 83.6% | 80.7% | 87.2% | Open |
| GPT-4o Mini | 66.5% | 92.4% | 91.3% | 83.4% | 81.1% | 86.5% | Closed |
| Gemma2 - 9B Instruct | 63.2% | 92.5% | 91.4% | 81.7% | 79.0% | 85.9% | Open |
| Gemini 1.5 Flash | 64.3% | 93.4% | 91.7% | 82.5% | 80.0% | 85.9% | Closed |
| Claude 3.5 Sonnet | 65.4% | 93.4% | 91.4% | 80.7% | 78.1% | 83.9% | Closed |
| Mixtral 8x22B | 64.3% | 91.2% | 90.1% | 79.2% | 75.2% | 83.0% | Open |
| Llama 3.1 8B Instruct Turbo | 62.6% | 90.4% | 86.1% | 77.8% | 73.9% | 80.1% | Open |

GPT-4o had significantly higher accuracy than all other LLMs - although the difference was non-significant for Llama 3.1 70B Instruct Turbo (OR = 1.118, z = 2.807, p = 0.134; S7 Table in S2 File). Llama 3.1 8B had significantly lower accuracy than all LLMs, whilst next worst was Mixtral 8x22 which had significantly lower accuracy than the rest of the LLMs (S7 Table in S2 File).

LLMs exhibited lower performance in closed book conditions compared to open book scenarios (Fig. 2; S3 Fig in S2 File; Table 2). When presented with only the correct document all LLMs showed high levels of performance (Table 2) with some, such as GPT-4o and Llama 3.1-70B, reaching 100% answer correctness across the human filtered questions (S8 Table in S2 File). Performance declined across all LLMs when an extraneous, irrelevant document was introduced in the 'confused' scenario as expected (S3 Fig in S2 File; Table 2).

**Retrieval accuracy across synopses, exam conditions, and retrieval strategies.** Retrieval accuracy significantly differed across both synopses and retrieval strategies (Fig. 3; S9 Table and S10 Table in S2 File). Across all LLMs, hybrid retrieval accuracy was significantly better than both dense and sparse retrieval (Table 3; S10 Table in S2 File, S11 Table in S2 File), as overall there was an increase in retrieving the correct document from 61.8% and 75.2% to 83.2% from sparse and dense to hybrid, respectively. This aligns with the increased accuracy of LLMs with the hybrid retrieval strategy compared to LLMs with dense or sparse retrieval strategies (Fig. 3; S3 Fig in S2 File; S6 Table in S2 File).

Across all synopses, hybrid retrieval accuracy was significantly greater than both dense (OR = 1.66, z = 6.17, p < 0.0001; S11 Table in S2 File) and sparse retrieval (OR = 3.14, z = 14.5, p < 0.0001; S11 Table in S2 File), as was found for LLM accuracy (Fig. 2; S6 Table in S2 File). Dense retrieval accuracy was also significantly greater than sparse retrieval (OR = 1.89, z = 8.75, p < 0.0001; S11 Table in S2 File). Dense retrieval resulted in the correct source document (used to generate the question) for 75.2% of questions, whilst this was only 61.8% for sparse retrieval (Table 3). Notably, in 85.8% of questions, dense retrieval returned at least one document from the same synopsis as the source document, for sparse this was 62.7% – only marginally higher than returning the correct source document.

For synopses, the poorest retrieval accuracy was found for the Mediterranean Farmland synopsis and significantly lower than more than half of the synopses (Fig. 3; S9 Table in S2 File). Several synopses (e.g., Marine and Freshwater Mammals, Marine Artificial Structures, Subtidal Benthic Invertebrates) had significantly higher retrieval accuracy than some of the synopses with the lowest retrieval accuracies (e.g., Reptile Conservation and Mediterranean Farmland; Fig 3; S12 Table in S2 File).

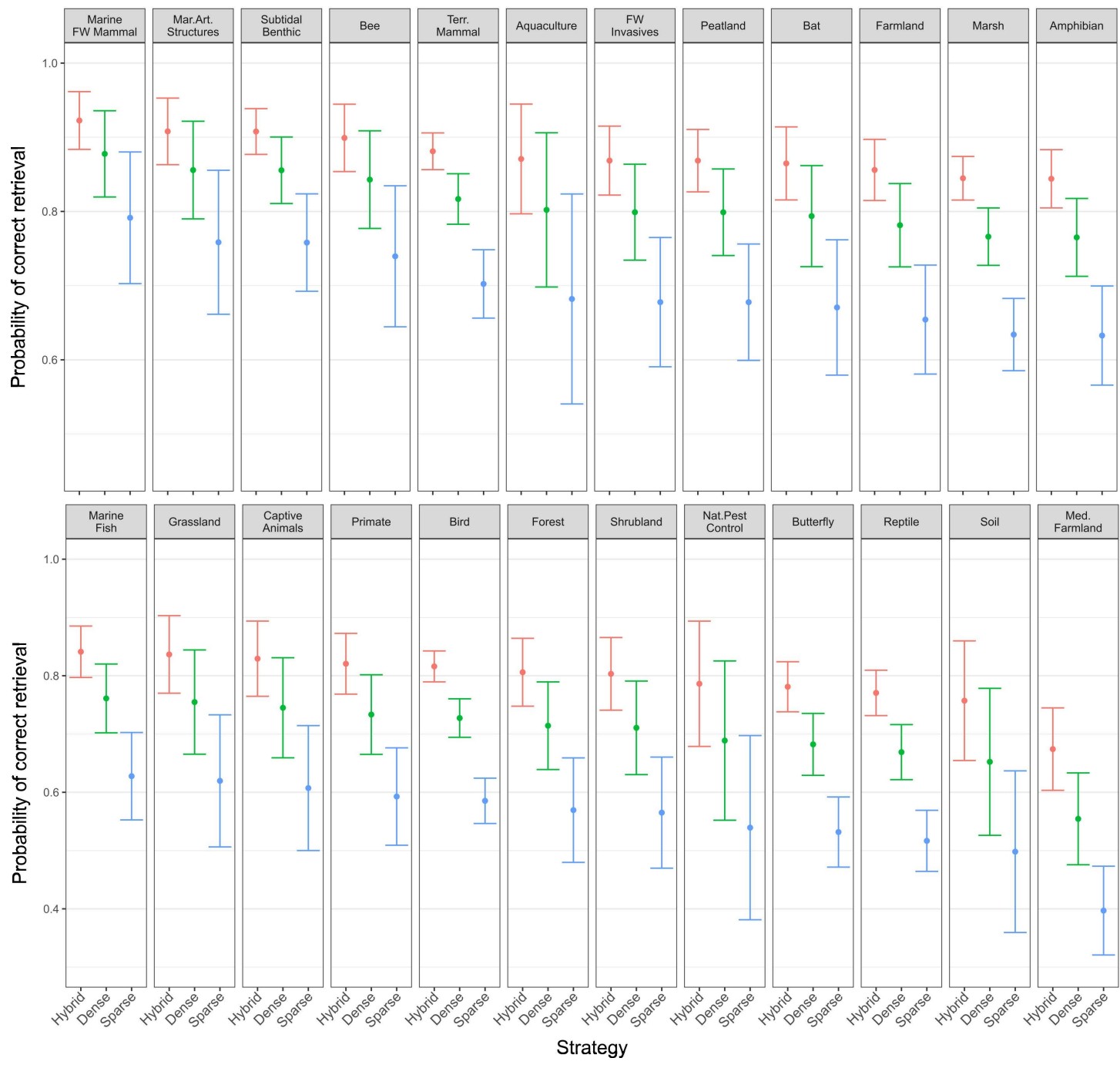

**Fig 3. Logistic regression Generalised Linear Model (GLM) predictions of the retrieval accuracy of different retrieval strategies across different synopses (mean and 95% Confidence Intervals).**

**Generational progress.** To assess the improvement of cutting-edge LLMs over time, we tested GPT-3.5 Turbo Instruct (released in September 2023 but derived from GPT-3.5 originally released in November 2022) on 679 unfiltered questions (S13 Table in S2 File). Performance in the oracle scenario was notably low for GPT-3.5 Turbo Instruct compared to all ten, more recent LLMs (Fig 2; Table 1).

**Table 3. Overall retrieval accuracy of different retrieval strategies across the unfiltered dataset of questions.**

| Measure | Retrieval strategy | | |
|---|---|---|---|
| | Dense | Sparse | Hybrid |
| Percentage of questions where source action document was retrieved correctly | 75.20% | 61.80% | 83.20% |
| Percentage of questions where document from same synopsis was retrieved (considers both selected documents) | 85.80% | 62.70% | 84.70% |
| Percentage of questions where source action document was selected first (if one of two documents selected was correct) | 80.00% | 82.60% | 78.20% |

## Discussion

Our key finding is that most LLMs using a hybrid retrieval strategy, in particular GPT-4o and Llama 3.1 70B Instruct Turbo, demonstrated comparable performance at answering multiple choice questions with a randomly selected human expert and far better than random guessing. In addition, retrieval accuracy for the hybrid strategy was also competitive with human expert retrieval. The retrieval performance was found to be lower than answer correctness for both LLMs and humans, likely due to overlapping key messages between actions, which enables related actions to inform responses to other questions. An example of this phenomenon is provided in S1 Fig in S2 File.

Our findings also suggest that the closed book performance of LLMs (without any retrieval strategy) demonstrated a level of conservation-specific knowledge, as evidenced by the fact that all LLMs performed better than random guessing under all exam conditions. This varied by topic area (Conservation Evidence synopses) but results remained high enough to suggest that the pre-training corpora incorporated relevant conservation literature – whilst many studies are published behind paywalls, the Conservation Evidence database website [18] and synopses are freely available online.

We also found that the performance of LLMs was consistent between the following exam conditions: oracle (providing the correct source action document) and confused (providing both a correct source action document with a random one). The confused exam condition is expected to outperform all other conditions (except oracle) since the correct action document is always provided (albeit with a random document, hence the term 'confusion'), whereas it is not in other scenarios, which should mean it is more likely to select the correct answer. Since retrieval in our study always results in two documents being provided to the LLM, the confused exam condition sets a ceiling for their performance – as expected, all LLMs have lower open book performance (using hybrid, dense, or sparse retrieval) than in oracle or confused performance across the larger unfiltered dataset. The hybrid retrieval strategy was found to perform substantially better than both dense and sparse retrieval strategies in terms of both question and retrieval accuracy, with dense also outperforming sparse. We also found that although the dense retrieval strategy's performance was higher when only considering whether it returned at least one document from the same synopsis as the source document, the sparse strategy's performance was negligibly higher. This is likely attributable to actions within the same synopsis being semantically similar and dense embeddings being able to capture this.

Finally, our findings also suggest there has been substantial generational improvement in LLM performance over a period of approximately two years, given that GPT-3.5 Turbo Instruct (derived from GPT-3.5 released in 2022) performed notably poorer than all ten more recently developed LLMs that we tested. This includes against models such as GPT-4o mini and Gemini 1.5 Flash which are considerably smaller and cheaper. Therefore, there does appear to be the potential for further improvements to the models we tested over the next few years, and emphasises the importance of our validation pipeline approach to rapidly test new models as they are released and calibrate them against expert-level training data.

## Limitations and future research

These findings need to be considered in the context of certain limitations. First, our results relate to LLMs' specific performance on the Conservation Evidence database, which may not generalise to questions on other conservation questions. For example, the database structure and nature of questions we asked about specific conservation actions meant that there was usually only one correct source document and it was unlikely another source document could be used to answer the question correctly – hence why our confused exam condition served the LLM two documents, one correct and one to 'confuse'. However, in other databases or use cases this may not be true and so future research may find it valuable to test whether there is a saturation point for performance in relation to the number of documents provided to the LLM (i.e., using more of the context window available and using different confused scenarios).

We also may not have captured the full range of questions that decision-makers have as the questions were derived from the Conservation Evidence database and generated by a LLM – thus any gaps and biases in the CE database and the LLM's pre-training data, including geographic, taxonomic, and language bias [33–35] may influence the questions set and thus the observed performance of LLMs. Nevertheless, we found that the observed performance of LLMs was broadly similar between the human-filtered questions and the larger dataset of unfiltered questions. However, LLMs are ultimately black box models, and we have a poor understanding of the pre-training data used to build them and their associated biases – thus further testing across different subject areas and domains remains important. Our results suggest that performance for certain synopses was poorer than others, although this did not necessarily follow an obvious taxonomic or biogeographical pattern. Our approach could be used to test LLM performance when retrieving information from databases, repositories and syntheses that already exist in other disciplines and topic areas – e.g., web databases such as the IUCN Red List and Education Endowment Foundation Toolkits that could be treated in a similar way to how the Conservation Evidence database was in our study.

It is also important to note that our evaluation method used Claude 3.5 Sonnet to generate the questions and answers and so all our questions should have been framed in a manner answerable by LLMs. However, Claude 3.5 Sonnet only performed in the middle to lower end of other LLMs tested, suggesting that it received no significant advantage in answering questions over the others. We must also acknowledge that despite setting the hyperparameter 'temperature' of all LLMs to zero, we cannot completely guarantee that LLM responses were 100% deterministic. However, whilst it is possible that LLMs may have provided a different response to a small number of questions if repeatedly prompted a large number of times, this is unlikely to have altered our major findings.

The human expert group we used was also limited in its sample size because we focused on testing LLMs against human experts that were highly familiar with navigating and answering questions using the Conservation Evidence database. However, this means that the human expert group we used is likely to represent an extremely high threshold of human performance compared to that of a more diverse group of experts – with general conservation expertise but less expertise on using the database or synthesising evidence on conservation interventions. In this context, however, the LLM performance we observed is even more impressive. We also suggest in the future that Item-Response Theory (IRT) is used to refine estimates of human expert versus LLM performance – for example, more extensive questions sets and larger, more diverse human groups could be used help to capture and account for variability in the difficulty of the questions and the participants' skill levels.

We may have also underestimated the retrieval times for human experts given that the questions we generated were based on a single conservation action and thus had a relatively high level of specificity – it may have taken experts longer to answer questions that rely on synthesising information across multiple conservation actions, and certainly would have taken a wider group of conservation experts without specific knowledge of the database more time too. Therefore, there are considerable potential time savings to be achieved by querying the Conservation Evidence database via a LLM-based RAG system over asking a human expert.

Finally, our exam-style evaluation scheme was designed to provide a plausible upper bound on LLMs' performance at answering questions, as a first step to evaluating their suitability for decision support when drawing upon evidence

databases and syntheses. There is likely to be more potential for errors when answering questions using free text generation as opposed to the constrained multiple-choice answers we examined. Furthermore, our evaluation only included a single task: answering a multiple-choice question on a single conservation action from a single synopsis. Now that we have demonstrated LLMs using a hybrid retrieval strategy are competitive with human experts at simple retrieval and question answering tasks, future research should investigate at which point LLM performance declines as tasks become more complex in terms of both questions and responses (e.g., questions requiring more synthesis across multiple actions or synopses with nuanced answers). Once this performance threshold with task complexity is identified, future systems should then be designed to stay above that threshold to avoid the risk of misinforming decision-makers. Our hypothesis would be that LLMs performance on more complex, nuanced questions and/or using free-text, open-ended answers would be lower than the results presented in our study, but it remains to be seen how much lower (if at all) and the types of errors that are made – for example, are they biased in a particular way, or do they provide overconfident or vague answers? Importantly, it is also crucial to understand whether these errors can be removed or mitigated against. For instance, there is also the potential to test whether more advanced prompt engineering can improve the performance of LLMs, as well as more computationally expensive embedding approaches that might improve retrieval performance.

**Risks**

It is also important to consider the ethics and clear associated risks of using LLMs in a decision support capacity to answer conservation questions based on evidence databases. First, equitable access to using LLMs for these purposes is important – any implementation should ideally be free and open access. This may also place an important constraint on which LLMs can be used in terms of cost – therefore, it is important to consider the relative cost-effectiveness of LLMs by comparing their performance to their cost. Indeed, we found that many open-source LLMs performed just as well as closed source LLMs (Table 2). Furthermore, access to LLMs will always be constrained, at least in part, by access to technology, expertise, power, and stable internet connections – although most decision support tools should be able to be run on relatively cheap devices, such as mobile phones. Therefore, the power dynamics between Global North and Global South institutions and organisations owning and using these tools should be carefully considered and recognised to avoid unequitable relationships at the research-practice and research-policy interfaces [36].

It is also important to recognise that the training of LLMs comes with substantial embodied environmental costs. These environmental costs have already occurred prior to the release of these models (i.e., in their initial training) and so there is arguably an ethical case that once developed LLMs should be used as much as possible for tasks that might mitigate these environmental costs. However, there are still costs associated with LLM inference via APIs (although these are a fraction of the original training costs) and so efficiency in the design of LLM-based RAG systems needs to be prioritised. Therefore, we should make sure that if there is a choice between using models with similar levels of performance, that we select the simpler, computationally cheaper models with lower environmental costs whenever possible.

There is also a risk that LLMs may lead to de-skilling of conservation practitioners and scientists in searching for, and assessing, the evidence. Although the goal is to make accessing and interpreting evidence databases and syntheses easier and more efficient, there is the risk that practitioners and scientists may think less critically about the evidence, its limitations, source, and validity. This could also lead to a lack of accountability in the use of evidence, including biased evidence use to support self-serving lines of argument and political rhetoric (e.g., by organisations for greenwashing). Therefore, it will be important to carefully design AI-assisted decision support systems to ensure they prompt users to understand the uncertainty and limitations associated with the evidence base, including its reliability and local relevance and transferability [5,16,33,37,38].

 

## Conclusion

We have shown that LLMs have comparable performance as human experts at providing restricted responses and retrieving relevant information to conservation intervention questions using the Conservation Evidence database. This first step in evaluating the performance of LLMs at decision support tasks now enables research to move on to establishing where any performance issues may occur when LLMs answer more complex, nuanced questions with open-ended, free-text responses. We stress that if performance issues are observed, we should design systems accordingly above this threshold to ensure the risk of misinforming decision-makers is minimised (with comparable or lower levels of error to human experts). For the Conservation Evidence database, our current findings suggest that it is justifiable to implement an LLM capable of providing an intelligent search function, to help direct users of the website to the most suitable page where evidence exists to answer their question. Our findings currently suggest that general LLMs used 'out-of-the-box' are likely to perform poorly at giving evidence-based advice or recommendations for decision-making. Any decisions based on such information are likely to be misinformed. Therefore, we urge those thinking of using general LLMs 'out-of-the-box' to provide decision support within their organisation or to end users, to spend time and effort to carefully consider how to responsibly design and evaluate LLM-based systems using retrieval augmented generation (RAG) and a hybrid retrieval strategy.

Looking to the future, if the rapid pace of improvements to underlying models continue, better performing LLMs coupled with further refinements of prompts and retrieval strategies could enable the development of RAG systems capable of expert-level, evidence-based advice using databases and syntheses. However, careful evaluation of LLM-based decision support systems is essential to ensure that more rapid and intuitive access to relevant evidence to inform practice and policy does not come at the cost of misinforming decision-makers with biased, erroneous, or misleading information.

## Supporting information

**S1 File. Participant information sheet and consent form.**
(DOCX)

**S2 File. A file containing all supporting figures and tables, including Figures S1-S3 and Tables S1-S13.**
(DOCX)

## Author contributions

**Conceptualization:** Radhika Iyer, Alec Philip Christie, Anil Madhavapeddy, Sam Reynolds, William Sutherland, Sadiq Jaffer.

**Data curation:** Radhika Iyer.

**Formal analysis:** Radhika Iyer, Alec Philip Christie, Sadiq Jaffer.

**Funding acquisition:** Alec Philip Christie, Anil Madhavapeddy, Sam Reynolds, William Sutherland, Sadiq Jaffer.

**Investigation:** Radhika Iyer, Alec Philip Christie, Sam Reynolds, Sadiq Jaffer.

**Methodology:** Radhika Iyer, Alec Philip Christie, Anil Madhavapeddy, Sam Reynolds, William Sutherland, Sadiq Jaffer.

**Project administration:** Alec Philip Christie, Sam Reynolds, Sadiq Jaffer.

**Resources:** Alec Philip Christie, Anil Madhavapeddy, Sam Reynolds, William Sutherland, Sadiq Jaffer.

**Software:** Radhika Iyer, Alec Philip Christie, Sadiq Jaffer.

**Supervision:** Alec Philip Christie, Sam Reynolds, William Sutherland, Sadiq Jaffer.

**Validation:** Radhika Iyer, Alec Philip Christie, Sadiq Jaffer.

**Visualization:** Radhika Iyer, Alec Philip Christie.

**Writing – original draft:** Radhika Iyer, Alec Philip Christie, Anil Madhavapeddy, Sam Reynolds, William Sutherland, Sadiq Jaffer.

**Writing – review & editing:** Radhika Iyer, Alec Philip Christie, Anil Madhavapeddy, Sam Reynolds, William Sutherland, Sadiq Jaffer.

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
