## [Decision Letter · Decision Letter 0]

11 Mar 2025

PONE-D-25-03431Careful design of Large Language Model pipelines enables expert-level retrieval of evidence-based information from syntheses and databasesPLOS ONE

Dear Dr. Christie,

Thank you for submitting your manuscript to PLOS ONE. After careful consideration, we feel that it has merit but does not fully meet PLOS ONE’s publication criteria as it currently stands. Therefore, we invite you to submit a revised version of the manuscript that addresses the points raised during the review process.The reviewers recognize the importance and timeliness of your work on Large Language Model (LLM) performance in retrieval strategies and expert comparison. However, they have identified several key areas that require revision before the manuscript can be considered for publication. Based on their feedback and editorial considerations, I am inviting you to submit a major revision of your manuscript.

Below, I summarize the primary concerns raised by the reviewers and editorial observations:

<h3>**Major Issues to Address:** </h3>

**Structural Clarity and Organization**The manuscript’s structure does not fully align with the standard PLOS ONE format. Methodological details are interspersed throughout the text rather than being consolidated within a distinct Methods section. Please revise the organization of your manuscript to ensure clarity, following the format: Introduction, Methods, Results, Discussion, and Conclusions.Additionally, a brief roadmap at the end of the Introduction outlining the structure of the paper would improve readability and guide the reader through the study’s logical progression.
**Methodological Enhancements and Justifications****Retrieval Strategy Justification:** Reviewer 2 notes that the "Confused" retrieval strategy, which introduces random documents, achieved the second-best performance, surpassing other strategies. A more in-depth discussion is needed on why this might be the case. If these additional documents provide beneficial information, explaining this phenomenon in greater detail would add clarity to your results.**Multiple Document Contexts:** In practical applications, users typically provide LLMs with multiple documents rather than just one. Reviewer 2 suggests expanding the context window by incorporating more retrieved documents (potentially 8-10) to evaluate whether performance improves or saturates at a certain point. This would provide a more rigorous and fair comparison with the "Confused" strategy.**Statistical Analysis:** The choice of statistical testing should be better justified. While your current approach (permutation test based on a sign test) is defensible, Reviewer 2 suggests McNemar’s test as a more straightforward and computationally efficient alternative. Please elaborate on your selection and consider whether McNemar’s test might be a more appropriate choice.
**Discussion of Generalizability and Future Research**Reviewer 1 suggests expanding the discussion on how LLMs perform when faced with open-ended, free-text responses rather than multiple-choice questions. Adding a short subsection or paragraph on this would enhance the study’s applicability to real-world decision-support scenarios.While you have outlined future research directions, strengthening this section with a discussion on item-response theory (IRT) would be beneficial. Emphasizing the potential for larger, more diverse human samples and broader question sets would improve the scalability of your findings.

<h3>**Minor Issues to Address:** </h3>

**Presentation and Formatting:**Box 1 is too large, spanning two pages and including a figure. Consider reformatting this section by presenting the figure separately with a caption to enhance readability.Table S4 contains variable names that are difficult to parse. Adding delimiters (e.g., underscores between words, such as “Model_gemini_flash” instead of “Modelgemini_flash”) would improve clarity.
<h3>**Next Steps:** </h3>Please submit your revised manuscript by Apr 25 2025 11:59PM. If you will need more time than this to complete your revisions, please reply to this message or contact the journal office at plosone@plos.org . Please include the following items when submitting your revised manuscript:

We look forward to receiving your revised manuscript.

Kind regards,

Carlos Carrasco-Farré

Academic Editor

PLOS ONE

Journal Requirements:

2. Thank you for stating the following financial disclosure: [RI was supported by a UROP internship at the University of Cambridge. APC received financial support from Imperial College London through an Imperial College Research Fellowship grant, as well as a Henslow Fellowship funded by the Cambridge Philosophical Society.]. 

4. Please include captions for your Supporting Information files at the end of your manuscript, and update any in-text citations to match accordingly. Please see our Supporting Information guidelines for more information: http://journals.plos.org/plosone/s/supporting-information .

Reviewers' comments:

Reviewer's Responses to Questions

**Comments to the Author**

1. Is the manuscript technically sound, and do the data support the conclusions?

Reviewer #1: Yes

Reviewer #2: Yes

2. Has the statistical analysis been performed appropriately and rigorously? 

Reviewer #1: Yes

Reviewer #2: I Don't Know

3. Have the authors made all data underlying the findings in their manuscript fully available?

Reviewer #1: Yes

Reviewer #2: Yes

4. Is the manuscript presented in an intelligible fashion and written in standard English?

Reviewer #1: Yes

Reviewer #2: Yes

5. Review Comments to the Author

Reviewer #1: One of the most beneficial additions would be a brief paragraph at the end of the Introduction describing how subsequent sections are organized. This “road map” can guide readers by clarifying the logical progression of your study—from context-setting in the introduction, to methodological details, through to results, discussion, and conclusions. Such a preview enables new readers to follow the paper’s narrative seamlessly.

While the use of multiple-choice questions provides a controlled environment for evaluation, real-world decision-making typically involves open-ended or more nuanced queries. Consider adding a separate paragraph or short subsection elaborating on how performance might differ when the LLMs are required to generate more complex, free-text answers. This discussion would give readers a clearer understanding of how generalizable your results are to practical, unconstrained decision support tasks.

Your study already outlines avenues for future research; however, you could strengthen this section by emphasizing the potential of item-response theory (IRT) for refining estimates of Large Language Model (LLM) vs. human performance. Discussing more extensive question sets with bigger and more diverse human samples would highlight the potential for capturing variability in question difficulty and participant skill levels. Such details would underscore the scalability of your approach and its capacity for broader, real-world application.

Reviewer #2: This paper studies the performance of LLMs across various retrieval strategies against human experts in answering synthetic multiple-choice questions on the effects of conservation interventions using the Conservation Evidence database. The authors performed extensive experiments and statistical tests, demonstrating that RAG systems can achieve the same level of performance as human experts.

Major Concerns:

1. While the paper compared 6 retrieval strategies and discussed them in the Discussion and Limitation sections, there is little explanation to why the “Confused” strategy achieves the 2nd best performance – only worse than “Oracle”. I understand the motivation is to introduce some random document and to see if the LLMs are confused, but apparently this random document does provide some extra, useful information. So if the authors can provide a little bit more in-depth analysis on this it would be great.

2. And speaking about providing extra documents, in practice, people actually feed the LLM a lot more than 1 document, because LLMs do have the ability to select the most relevant information from a relatively larger context than just 1 document. And even if there is just 1 “golden” document, other documents may also provide similar information. Typically the more documents you include in the context, the better the performance. And this effect only starts to saturate starting at 4k tokens (Cf. https://www.databricks.com/blog/long-context-rag-performance-llms), which means around 8-10 documents, estimated based on the length of documents you provided in the supplementary. And by doing this, I think you can show more rigorous experiment results and make a fair comparison with the “Confused” strategy since now you can have an equal number of retrieved documents.

3. Can you elaborate more on why you chose this specific statistical testing approach (something like a permutation test based on a sign test)? I think it does make some sense on dealing with “draw by chance”, but wouldn’t the McNemar’s test be more straightforward and efficient, and less computationally expensive?

Minor Issues:

1. I’m not a fan of the idea putting a lot of things in a big box (Box 1). The box is too big, crossing two pages, and contains a figure, which could be presented separately with a proper caption.

2. The variable names in Table S4 is hard to read since there is no delimiter between the type of variable and the actual name (e.g., “Modelgemini_flash”). It could be much better by just adding a underscore in between (e.g., “Model_gemini_flash”).

6. PLOS authors have the option to publish the peer review history of their article (what does this mean? ). If published, this will include your full peer review and any attached files.

**Do you want your identity to be public for this peer review?** For information about this choice, including consent withdrawal, please see our Privacy Policy .

Reviewer #1: No

Reviewer #2: No

---

## [Author Response · Author response to Decision Letter 1]

7 Apr 2025

PONE-D-25-03431

Careful design of Large Language Model pipelines enables expert-level retrieval of evidence-based information from syntheses and databases

Editor’s comments:

Thank you for submitting your manuscript to PLOS ONE. After careful consideration, we feel that it has merit but does not fully meet PLOS ONE’s publication criteria as it currently stands. Therefore, we invite you to submit a revised version of the manuscript that addresses the points raised during the review process.

The reviewers recognize the importance and timeliness of your work on Large Language Model (LLM) performance in retrieval strategies and expert comparison. However, they have identified several key areas that require revision before the manuscript can be considered for publication. Based on their feedback and editorial considerations, I am inviting you to submit a major revision of your manuscript.

Response: We thank the editor and reviewers for their comments and agree that the suggested revisions have helped to strengthen the manuscript. We hope the changes made now make the publication suitable for publication.

Below, I summarize the primary concerns raised by the reviewers and editorial observations:

Major Issues to Address:

1. Structural Clarity and Organization

o The manuscript’s structure does not fully align with the standard PLOS ONE format. Methodological details are interspersed throughout the text rather than being consolidated within a distinct Methods section. Please revise the organization of your manuscript to ensure clarity, following the format: Introduction, Methods, Results, Discussion, and Conclusions.

Response: We have now adjusted the manuscript structure to align with PLOS ONE format.

o Additionally, a brief roadmap at the end of the Introduction outlining the structure of the paper would improve readability and guide the reader through the study’s logical progression.

Response: We have edited the last paragraphs of the introduction to do this (L123-158) and moved the figure out of Box 1 to here (now Figure 1).

2. Methodological Enhancements and Justifications

o Retrieval Strategy Justification: Reviewer 2 notes that the "Confused" retrieval strategy, which introduces random documents, achieved the second-best performance, surpassing other strategies. A more in-depth discussion is needed on why this might be the case. If these additional documents provide beneficial information, explaining this phenomenon in greater detail would add clarity to your results.

Response: We have added more clarity on this in the main text (Methods L296-326, Discussion L574-580) and respond in detail to the reviewer’s comment below.

o Multiple Document Contexts: In practical applications, users typically provide LLMs with multiple documents rather than just one. Reviewer 2 suggests expanding the context window by incorporating more retrieved documents (potentially 8-10) to evaluate whether performance improves or saturates at a certain point. This would provide a more rigorous and fair comparison with the "Confused" strategy.

Response: As per our previous comment, we have added more clarity on this in the main text and respond to the reviewer’s comment below (Discussion L605-612).

o Statistical Analysis: The choice of statistical testing should be better justified. While your current approach (permutation test based on a sign test) is defensible, Reviewer 2 suggests McNemar’s test as a more straightforward and computationally efficient alternative. Please elaborate on your selection and consider whether McNemar’s test might be a more appropriate choice.

Response: We now better justify our approach and respond directly to the reviewer’s comment below (Methods L358-366).

3. Discussion of Generalizability and Future Research

o Reviewer 1 suggests expanding the discussion on how LLMs perform when faced with open-ended, free-text responses rather than multiple-choice questions. Adding a short subsection or paragraph on this would enhance the study’s applicability to real-world decision-support scenarios.

Response: We have now elaborated on this point and responded to the reviewer’s comment below (Discussion L676-684).

o While you have outlined future research directions, strengthening this section with a discussion on item-response theory (IRT) would be beneficial. Emphasizing the potential for larger, more diverse human samples and broader question sets would improve the scalability of your findings.

Response: We agree and have elaborated on IRT where suggested by the reviewer (Discussion L650-654).

Minor Issues to Address:

• Presentation and Formatting:

o Box 1 is too large, spanning two pages and including a figure. Consider reformatting this section by presenting the figure separately with a caption to enhance readability.

Response: We understand this and have now separated the Figure from Box 1 and left Box 1 as a shortened glossary. The figure is now Figure 1 and referred to from Box 1. We think there is still value in having a more concise Box 1 given the need to explain various terms upfront in the paper.

o Table S4 contains variable names that are difficult to parse. Adding delimiters (e.g., underscores between words, such as “Model_gemini_flash” instead of “Modelgemini_flash”) would improve clarity.

Response: We have added underscores as suggested through the Supporting Information.

Journal Requirements:

Response: We have now edited the document to comply with these style requirements to the best of our knowledge.

2. Thank you for stating the following financial disclosure: [RI was supported by a UROP internship at the University of Cambridge. APC received financial support from Imperial College London through an Imperial College Research Fellowship grant, as well as a Henslow Fellowship funded by the Cambridge Philosophical Society.].

Response: Thank you. Our financial disclosure now states: “RI was supported by a UROP internship at the University of Cambridge. APC received financial support from Imperial College London through an Imperial College Research Fellowship grant, as well as a Henslow Fellowship funded by the Cambridge Philosophical Society. The funders had no role in study design, data collection and analysis, decision to publish, or preparation of the manuscript.”

Response: I can confirm that the ethics statement is in the methods section (L238-244).

Response: We have now provided these in the main text after the Reference list.

Reviewer #1 Comments:

Comment:

One of the most beneficial additions would be a brief paragraph at the end of the Introduction describing how subsequent sections are organized. This “road map” can guide readers by clarifying the logical progression of your study—from context-setting in the introduction, to methodological details, through to results, discussion, and conclusions. Such a preview enables new readers to follow the paper’s narrative seamlessly.

Response: We have edited the last paragraphs of the introduction to do this (L123-158) and moved the figure out of Box 1 to here (now Figure 1).

Comment:

While the use of multiple-choice questions provides a controlled environment for evaluation, real-world decision-making typically involves open-ended or more nuanced queries. Consider adding a separate paragraph or short subsection elaborating on how performance might differ when the LLMs are required to generate more complex, free-text answers. This discussion would give readers a clearer understanding of how generalizable your results are to practical, unconstrained decision support tasks.

Response: This was a useful point and we have now elaborated on how LLMs performance might differ with open-ended, free-text response evaluations that we plan to do in the future (Discussion L676-684).

Comment:

Your study already outlines avenues for future research; however, you could strengthen this section by emphasizing the potential of item-response theory (IRT) for refining estimates of Large Language Model (LLM) vs. human performance. Discussing more extensive question sets with bigger and more diverse human samples would highlight the potential for capturing variability in question difficulty and participant skill levels. Such details would underscore the scalability of your approach and its capacity for broader, real-world application.

Response: We agree and have elaborated on IRT where suggested by the reviewer (Discussion L650-654).

Reviewer #2 Comments:

Comment:

This paper studies the performance of LLMs across various retrieval strategies against human experts in answering synthetic multiple-choice questions on the effects of conservation interventions using the Conservation Evidence database. The authors performed extensive experiments and statistical tests, demonstrating that RAG systems can achieve the same level of performance as human experts.

Major Concerns:

1. While the paper compared 6 retrieval strategies and discussed them in the Discussion and Limitation sections, there is little explanation to why the “Confused” strategy achieves the 2nd best performance – only worse than “Oracle”. I understand the motivation is to introduce some random document and to see if the LLMs are confused, but apparently this random document does provide some extra, useful information. So if the authors can provide a little bit more in-depth analysis on this it would be great.

Response: We understand where the confusion might arise relating to the confused scenario. The key is that the confused scenario always include the correct source document and therefore is most closely related to the oracle scenario (which only provides the source document). Therefore, despite there being a potentially ‘confusing’ document present, it is more likely that the LLM is able to answer the question correctly than the other scenarios because the correct source document is always present (whereas that is not the case for scenarios other than oracle and confused). We have added more clarity on this in the main text (Methods L296-326, Discussion L574-580).

Comment:

2. And speaking about providing extra documents, in practice, people actually feed the LLM a lot more than 1 document, because LLMs do have the ability to select the most relevant information from a relatively larger context than just 1 document. And even if there is just 1 “golden” document, other documents may also provide similar information. Typically the more documents you include in the context, the better the performance. And this effect only starts to saturate starting at 4k tokens (Cf. https://www.databricks.com/blog/long-context-rag-performance-llms), which means around 8-10 documents, estimated based on the length of documents you provided in the supplementary. And by doing this, I think you can show more rigorous experiment results and make a fair comparison with the “Confused” strategy since now you can have an equal number of retrieved documents.

Response: This is a great point generally about LLMs, but the way the Conservation Evidence database is structured and the way the questions were generated, the answers to the questions are typically contained in a single document, rather than across multiple, certainly no more than two (e.g., two similar conservation actions – most actions are quite specific) in the vast majority of circumstances – i.e., the answer to a question usually only requires a single source document to answer it, given it is a written summary of the evidence for a conservation action and the question being asked relates to a specific conservation action. Therefore, there would be no advantage to selecting more than two documents – and in fact, this is likely to result in poorer performance and represent an inefficient use of computation. However, this is an important point that we have decided to discuss in the discussion alongside Reviewer #1’s comments relating to future research to evaluate LLM performance on nuanced questions with free-text responses (Discussion L605-612).

Comment:

3. Can you elaborate more on why you chose this specific statistical testing approach (something like a permutation test based on a sign test)? I think it does make some sense on dealing with “draw by chance”, but wouldn’t the McNemar’s test be more straightforward and efficient, and less computationally expensive?

Response: We did consider McNemar’s test but we wanted to account for the fact that: 1. We need to adjust for the probability that humans and LLMs give the same answer by chance as they were answering multiple choice questions with four possible responses, which McNemar’s test does not account for; 2. We are also specifically interested in cases where there are draws, an LLM outperforms a random human, and where a random human outperforms an LLM; 3. Our sample size was limited by the need for a comparison with a specific group of human experts and therefore our bootstrapped approach enabled a more robust comparison than had we just used McNemar’s test. We have added justification to this effect into the main text (Methods L358-366).

Comment:

Minor Issues:

1. I’m not a fan of the idea putting a lot of things in a big box (Box 1). The box is too big, crossing two pages, and contains a figure, which could be presented separately with a proper caption.

Response: We understand this and have now separated the Figure from Box 1 and left Box 1 as a shortened glossary. The figure is now Figure 1 and referred to from Box 1. We think there is still value in having a more concise Box 1 given the need to explain various terms upfront in the paper.

Comment:

2. The variable names in Table S4 is hard to read since there is no delimiter between the type of variable and the actual name (e.g., “Modelgemini_flash”). It could be much better by just adding a underscore in between (e.g., “Model_gemini_flash”).

Response: We have added underscores as suggested through the Supporting Information.

---

## [Decision Letter · Decision Letter 1]

10 Apr 2025

Careful design of Large Language Model pipelines enables expert-level retrieval of evidence-based information from syntheses and databases

PONE-D-25-03431R1

Dear Dr. Christie,

We’re pleased to inform you that your manuscript has been judged scientifically suitable for publication and will be formally accepted for publication once it meets all outstanding technical requirements.

Kind regards,

Carlos Carrasco-Farré

Academic Editor

PLOS ONE

Additional Editor Comments (optional):

Following reviewer's 1 comments, we recommend adding a brief paragraph at the end of the introduction section showing the layout of the paper. E.g., In the next section (section 1) we present a case study explaining experimental/data setup followed by section 2 discussing X .... and so on.

Reviewers' comments:

Reviewer's Responses to Questions

**Comments to the Author**

1. If the authors have adequately addressed your comments raised in a previous round of review and you feel that this manuscript is now acceptable for publication, you may indicate that here to bypass the “Comments to the Author” section, enter your conflict of interest statement in the “Confidential to Editor” section, and submit your "Accept" recommendation.

Reviewer #1: All comments have been addressed

Reviewer #2: All comments have been addressed

2. Is the manuscript technically sound, and do the data support the conclusions?

Reviewer #1: Partly

Reviewer #2: Yes

3. Has the statistical analysis been performed appropriately and rigorously? 

Reviewer #1: Yes

Reviewer #2: Yes

4. Have the authors made all data underlying the findings in their manuscript fully available?

Reviewer #1: Yes

Reviewer #2: Yes

5. Is the manuscript presented in an intelligible fashion and written in standard English?

Reviewer #1: Yes

Reviewer #2: Yes

6. Review Comments to the Author

Reviewer #1: Please include links / references in as citations e.g., (www.conservationevidence.com)

I would still question the writing style used here, I am all for creativity but this writing style is unconventional for the field. I recommend adding a brief para at the end of the introduction section showing the layout of the paper. E.g., In the next section (section 1) we present a case study explaining experimental/data setup followed by section 2 discussing X .... and so on.

Reviewer #2: (No Response)

7. PLOS authors have the option to publish the peer review history of their article (what does this mean? ). If published, this will include your full peer review and any attached files.

**Do you want your identity to be public for this peer review?** For information about this choice, including consent withdrawal, please see our Privacy Policy .

Reviewer #1: No

Reviewer #2: No

---

## [Editor Report · Acceptance letter]

PONE-D-25-03431R1

PLOS ONE

Dear Dr. Christie,

I'm pleased to inform you that your manuscript has been deemed suitable for publication in PLOS ONE. Congratulations! Your manuscript is now being handed over to our production team.

Kind regards,

on behalf of

Dr. Carlos Carrasco-Farré

Academic Editor

PLOS ONE